# Synthesis and Characterisation of Platinum(II) Diaminocyclohexane Complexes with Pyridine Derivatives as Anticancer Agents

**DOI:** 10.3390/ijms242417150

**Published:** 2023-12-05

**Authors:** Brondwyn S. McGhie, Jennette Sakoff, Jayne Gilbert, Christopher P. Gordon, Janice R. Aldrich-Wright

**Affiliations:** 1Nanoscale Organisation and Dynamics Group, School of Science, Western Sydney University, Locked Bag 1797, Penrith, NSW 2751, Australia; brondwyn.mcghie@westernsydney.edu.au (B.S.M.); c.gordon@westernsydney.edu.au (C.P.G.); 2Department of Medical Oncology, Calvary Mater Newcastle Hospital, Waratah, NSW 2298, Australia; jennette.sakoff@newcastle.edu.au (J.S.); jayne.gilbert@newcastle.edu.au (J.G.)

**Keywords:** anticancer drugs, platinum complexes, imidazole, pyridine derivatives

## Abstract

Cisplatin-type covalent chemotherapeutics are a cornerstone of modern medicinal oncology. However, these drugs remain encumbered with dose-limiting side effects and are susceptible to innate and acquired resistance. The bulk of platinum anticancer research has focused on Cisplatin and its derivatives. Here, we take inspiration from the design of platinum complexes and ligands used successfully with other metals to create six novel complexes. Herein, the synthesis, characterization, DNA binding affinities, and lipophilicity of a series of non-traditional organometallic Pt(II)-complexes are described. These complexes have a basic [Pt(P_L_)(A_L_)]Cl_2_ molecular formula which incorporates either 2-pyrrolidin-2-ylpyridine, 2-(1*H*-Imidazol-2-yl)pyridine, or 2-(2-pyridyl)benzimidazole as the P_L_; the A_L_ is resolved diaminocyclohexane. Precursor [Pt(P_L_)(Cl)_2_] complexes were also characterized for comparison. While the cytotoxicity and DNA binding properties of the three precursors were unexceptional, the corresponding [Pt(P_L_)(A_L_)]^2+^ complexes were promising; they exhibited different DNA binding interactions compared with Cisplatin but with similar, if not slightly better, cytotoxicity results. Complexes with 2-pyrrolidin-2-ylpyridine or 2-(2-pyridyl)benzimidazole ligands had similar DNA binding properties to those with 2-(1*H*-Imidazol-2-yl)pyridine ligands but were not as cytotoxic to all cell lines. The variation in activity between cell lines was remarkable and resulted in significant selectivity indices in MCF10A and MCF-7 breast cancer cell lines, compared with previously described similar Pt(II) complexes such as 56MESS.

## 1. Introduction

Cancer is rapidly becoming the leading cause of premature death worldwide, overtaking heart disease, particularly in nations with low to moderate Human Development Index scores [1,2,3]. While prevention may reduce the burden of cancer by up to 50% [3,4], safer, more effective, and affordable treatment options are required to reduce premature death in line with the United Nations Sustainable Development Goal (SDG) [4]. Improved therapies are necessary to address the challenges faced by advanced-stage cancer patients who often receive poor prognoses [5,6,7]. The burden of cancer has significant repercussions on both the broader societal and economic landscapes, even in cases where cancer enters remission [8,9]. Platinum(II)-based chemotherapy drugs, most commonly Cisplatin, Oxaliplatin, and Carboplatin, are used globally. However, many cancers are resistant to these drugs, and they have significant side effects, which can necessitate reduced doses, causing suboptimal outcomes [1,10,11]. Therefore, the search for a “cure for cancer” is still very much underway. Inorganic chemists aspire to create more potent, effective, and cancer cell-selective metal complexes to achieve better patient prognoses [12,13].

Prior research studies by our group have centered on non-traditional structures of platinum complexes (PCs) with promising results [14,15]. Specifically, the example [(5,6-dimethyl-1,10-phenanthroline)(1*S*,2*S*-diaminocyclohexane)platinum(II)] dichloride (56MESS) has received warranted research interest [16,17,18,19]. This non-traditional PC is of the type [Pt(P_L_)(A_L_)]^2+^ where P_L_ is a polyaromatic ligand and A_L_ is an ancillary ligand, usually diaminocyclohexane. The unique design of 56MESS and similar complexes has resulted in their mechanism being distinct from Cisplatin. Moreover, 56MESS and its derivatives elicit anti-proliferative effects against Cisplatin-resistant KRAS-mutated cells [20,21]. Many of these examples are significantly more potent than Cisplatin in 2D cell culture studies, have increased stability when oxidized to platinum(IV), and have improved selectivity indices in breast cancer cell lines [15,22].

In this study, we present a new series of unconventional PCs and assess their anticancer potential. These complexes exhibit a similar structure to 56MESS, [Pt(P_L_)(A_L_)]^2+^; however, the P_L_ is a pyridine derivative, while the A_L_ remains as chiral resolved diaminocyclohexane (Figure 1). The ligands investigated here have been used previously to probe the anticancer activity of Cu, Ru, Os, Pt, and Pd complexes [23,24,25,26]. These ligands, however, are yet to be used in a Pt(II) complex context where the pyridine ligand is bidentate and has a second chiral diaminocyclohexane ligand. It is hypothesized that pyridine derivatives influence complex solubility and cell permeability as well as affect reactivity toward different biomolecules through hydrogen bonding, π-π stacking, and intercalation [25,26,27]. The antihistamine, antibacterial, antiviral, antifungal, and anti-diabetic utility of these ligands has been investigated for many pharmaceutical applications, in addition to their cytotoxicity toward cancer cells [23,24,25,27,28,29,30].

The synthesis and characterization of the six novel and three precursor complexes are described herein. HPLC, UV, NMR, and ESI-MS indicate that the complexes were synthesized at good yields and purity. The complexes **4**–**6a**/**b** retain the chirality of the diaminocyclohexane (DACH) used in their synthesis, and this was confirmed using circular dichroism (CD). The suitability of **4**–**6a**/**b** and their precursors **1**–**3** for use as anticancer agents was evaluated by measuring their cytotoxicity, DNA binding, and lipophilicity.

## 2. Results and Discussion

### 2.1. Synthesis

The synthetic strategy used was based on a recently published method whereby the poly-heteroaromatic ligand is first coordinated to the Pt(II) center before the coordination of the ancillary DACH ligand (Figure 1) [22]. This contrasts with the strategy used in the synthesis of other non-conventional PCs mentioned previously, where the Pt-DACH complex is synthesized first. This method resulted in good purity and yield and was advantageous as it produced complexes **1**–**3**, which could then be used as test complexes to assess the influence of the chiral ligand. The addition of 2-methoxy ethanol proved highly effectual for the dissolution of the starting materials and subsequent coordination to platinum; however, if a 1:5 ratio with water was exceeded, the PC failed to precipitate. In this event, chromatographic isolation was required and resulted in significantly reduced product returns. Additionally, the coordination of the DACH ligand was heat sensitive; at reaction temperatures > 95 °C, the platinum would reduce to Pt(0), resulting in diminished yields. The ideal temperature for this step was found to be 80 °C. The complexes were observed to be stable in solution over extended periods; clean HPLC spectra could be obtained weeks after dilution in DMSO or H_2_O (in Appendix A).

### 2.2. Chemical Characterisation

Each PC (**4**–**6a**/**b**) and their starting material (**1**–**3**) were characterized using a combination of NMR spectroscopy and HPLC. The PCs **4**–**6a**/**b** were further characterized by circular dichroism (CD), UV spectroscopy, and ESI-MS (Table 1). The purity was determined to be greater than 95% by HPLC (Appendix A). The CD spectra confirmed that the chirality of the starting materials was retained during synthesis. Additionally, the chiral purity of the complexes was confirmed, with the SS and RR (a/b) spectra being almost perfect mirror images (Figure 2 and Appendix A). The correct mass peak was identified in all samples using ESI-MS at 2+ ion *m*/*z*. All MS data are presented in the Appendix A.

The NMR characterizations of **1**–**6** were achieved using a combination of ^1^H, ^195^Pt, ^1^H-^195^Pt heteronuclear multiple quantum correlations (HMQC), and ^1^H-^1^H COSY spectra. All experiments were conducted in DMSO to maintain consistency between all complexes, as **1**–**3** were only soluble in H_2_O at low concentrations. The acquired NMR spectra were consistent with the expected values. These spectra are provided in full in the Appendix A. The NMR spectra showed successful coordination of each pyridine ligand and subsequent addition of DACH. There were minimal differences between the **a**/**b** variants of complexes **4**–**6**. However, altering the concentration of the PC resulted in the slight shifting of the peaks in the aromatic region. This was due to π-π stacking of the pyridine ligand. The ^195^Pt spectra of the starting material (**1**–**3**) and that of the **4**–**6** complexes were as expected from the literature [31]. However, small shifts (~200 ppm) were significant enough to distinguish between the coordination sphere of each intermediate, Pt(Cl)_4_:Pt(N)_2_(Cl)_2_:Pt(N), respectively. The aliphatic region was significantly split compared with the spectra obtained for 56MESS-type complexes, including asymmetric derivatives (Figure 3). The asymmetry of the pyridine-derived ligands is likely having a greater impact on the electrochemistry of the complex due to the imidazole’s increased electron-donating properties compared with pyridine. Furthermore, the ^1^H-^195^Pt HMQC spectra showed additional peaks; protons further from the platinum center had strong correlations with up to four protons in the aromatic region. This suggests that these products, although square planar, are not as flat as phenanthroline-based complexes, although no crystals were able to be grown to verify this (Figure 4).

UV spectra showed minimal variation between **a**/**b** complexes, which is unsurprising given that the polypyridyl ligand is the determining factor on the UV spectra of all previous Pt(II) 56MESS derivatives. Further, UV data were used to calculate the molar absorption coefficient for each complex at two wavelengths. Molar absorption coefficients were calculated by titrating increasing concentrations of a complex and measuring the UV spectra of each increment. The six complexes have relatively low absorption in water and few real peaks in their spectra. Despite this, the extinction coefficients were successfully calculated with good agreement between the three repetitions (Appendix A).

Lipophilicity was calculated using RP-HPLC. A stock solution of each PC was injected at different isocratic ratios ranging from 70–90% solvent B (organic) at a flow rate of 1 mL·min^−1^ [32,33,34]. The complexes have low solubility in water, often requiring DMSO to make concentrated solutions. Yet the LogK’ calculated using these experimental data indicated high lipophilicity, which is typical for 2+ charged complexes. Further experimentation found that these complexes show only an initial resistance to dissolution; they require time or heat to dissolve fully. When heat was applied, they would not precipitate out of the solution after cooling. This confirmed that a lack of persistence was the cause of the “observed” low solubility in H_2_O. This suggests that these complexes have low waters of hydration when synthesized using the above method and thus take longer to dissolve than previous 56MESS-derived complexes that typically had high waters of hydration. In the context of Pt(II) anticancer agents, the complexes noted in this study are more lipophilic than comparable drugs in the literature. Their increased lipophilicity may indicate they have the potential to be administered orally, a characteristic usually associated with Pt(IV) complexes. Other factors, such as stability and cellular absorption, would need to be considered before the potential of these Pt(II) anticancer agents could be evaluated.

### 2.3. Biophysical Characterisation

The DNA binding capacity of PCs **1**–**6** was assessed using fluorescent intercalator displacement (FID) experiments. To achieve this, the fluorescent signal of a DNA intercalator, in this case, ethidium bromide (EtBr), is measured as increasing concentrations of PC are added. As the PC displaces the EtBr, the fluorescence of the mixture is diminished, and the DNA binding capacity of the complex can be calculated from the fluorescence measurement. Calf-thymus DNA (ctDNA) was used and saturated with EtBr. The change in fluorescence is monitored as the complex is titrated into this solution. Once the decrease in fluorescence plateaued, the titration was ended. Experiments were performed in triplicate for each PC, and the DNA binding capacity is summarised in Table 2.

The results show that greater molar equivalence is needed to displace EtBr for complexes **1**–**3** than for **4**–**6**. The fluorescence plateaued using fewer equivalents for **4**–**6** and resulted in a greater total change in fluorescence, indicating that these PCs were able to displace more EtBr than precursors **1**–**3**. Therefore, the PCs are much stronger binders, as is additionally evident through the K_a_ and K_SV_ values.

The cytotoxicity of **1**–**6** was assessed using MTT assays that were undertaken in several different cell lines, including a non-cancerous cell line. The resulting GI_50_ values (Table 3) demonstrated that **5a**/**b** and **6a** have a similar if slightly improved, activity to Cisplatin in all cell lines tested. Surprisingly, the *RR* complex **6b** did not have the same activity, with GI_50_ values increased by a factor of ten compared with its *SS* counterpart, **6a**. Precursor complexes **1**–**3** did not have impressive cytotoxicity values, although they are somewhat comparable with the cytotoxicity of Carboplatin. Their development may not be worth pursuing further due to their similarity to conventional Pt(II) complexes, which are known to be cross-resistant. Complexes **4a**/**b** were also disappointing, with GI_50_ values in a similar range to the precursor complexes. Interestingly, **4a**/**b** showed the greatest variation in activity between the 12 cell lines, suggesting that these could be used to create more targeted complexes. Complexes **4a**/**b** and **6b** also have higher selectivity indices than **5a**/**b** and **6a**, which further indicates their potential as useful tools for targeted therapies. In comparison, published anticancer agents containing similar ligands have not achieved such low GI_50_ values. For example, ruthenium complexes with piano chair benzimidazole motifs produced, at best, GI_50_ values of 11–19 µM in MCF-7 cells and 26–38 µM in A2780 cells [27]. Platinum complexes that similarly coordinate imidazole produced modest cytotoxicity with the best results between 17.5 and 58.8 µM in the cell lines tested, although no direct cell line comparisons were available [24,28]. The results of complexes from the literature are comparable with precursors **1**–**3** and less effective compared with complexes **4**–**6**. We can, therefore, confirm that the design has improved the cytotoxicity of this family of ligands, and this improvement validates our drug design. This may be due to the increased lipophilicity of these complexes compared with other platinum derivatives or because they are designed for intercalation rather than covalent DNA binding.

The cytotoxicity correlates with the DNA binding affinity where **1**–**3** have the lowest DNA binding capacity and are less cytotoxic compared with complexes **4**–**6**. Complex **6b** is the exception, which, although it has moderate DNA binding properties, was significantly less cytotoxic. This suggests that DNA binding may contribute to the mechanism of action of these PCs, although other properties may hinder DNA binding capacity in vitro, thus reducing their cytotoxicity. Overall, the binding and cytotoxicity results show PCs **5**–**6** are the most promising and can be used as a scaffold for Pt(IV) prodrugs, while PCs **4a**/**b** may be best utilized in Pt(IV) drugs designed for targeting to utilize their significantly increased potency in breast, brain and pancreatic cancer cells compared with the other cell lines tested.

## 3. Materials and Methods

### 3.1. Materials

Reagents were used as received. All solvents used were of analytical grade or higher and purchased from Labserv (Edwardstown, Australia), Chem-Supply (Gillman, Australia), or Merck Chemicals (North Ryde, Australia). Potassium tetrachloroplatinate (K_2_PtCl_4_) was purchased from Precious Metals Online (Wollongong, Australia). Trifluoroacetic acid (TFA), 2-Pyrrolidin-2-ylpyridine, 2-(1*H*-Imidazol-2-yl)pyridine, and 2-(2-pyridyl)benzimidazole were purchased from Sigma-Aldrich (Bayswater, Australia). Calf thymus DNA was purchased from ThermoFisher (Parkville, Australia) and cell lines were purchased from ATCC Scientific (Manassas, VA, USA). Methanol, acetonitrile (ACN), ethanol, and methoxyethanol were obtained from Honeywell. Deuterated solvent *d*_6_-dimethylsulphoxide (DMSO-*d*_6_, 99.9%) was purchased from Cambridge Isotope Laboratories.

### 3.2. Synthesis

#### 3.2.1. Synthesis of [Pt(PyPy)(Cl)_2_] (**1**), [Pt(ImPy)(Cl)_2_] (**2**) and [Pt(BImPy)(Cl)_2_] (**3**)

Potassium tetrachloroplatinate (1 equiv.) was dissolved in a 5:1 solution of water and methoxyethanol. Then 2-pyrrolidin-2-ylpyridine (PyPy) (1.1 equiv.) was added to the solution before being stirred at 40 °C for 4 h. The solution was then left to cool, allowing [Pt(PyPy)(Cl)_2_] (**1**) to precipitate. The solution was then filtered, and the product was washed with ~5 mL of diethylether. This method was repeated using 2-(1*H*-Imidazol-2-yl)pyridine (ImPy), and 2-(2-pyridyl)benzimidazole (BImPy) as the ligand to produce [Pt(ImPy)(Cl)_2_] (**2**) and [Pt(BImPy)(Cl)_2_] (**3**). **1** ^1^H NMR: δ 9.15(d, *J* = 5.85 Hz, H1), δ 7.49(t, *J* = 6.88 Hz, H1), δ 8.16(t, *J* = 7.78 Hz, H1), δ 7.59(d, *J* = 7.93 Hz, H1), δ 4.60(dd, *J* = 15.24, 7.71 Hz, H1), δ 2.47(m, H merged with solvent peak), δ 1.86(m, H2), δ 3.09(m, H2), ^195^Pt: −3115.8 ppm. **2** ^1^H NMR: δ 9.28(d, *J* = 5.74 Hz, H1), δ 7.81(m, H1), δ 8.34(t, *J* = 7.75 Hz, H1), δ 8.15(d, *J* = 8.15 Hz, H1), δ 7.37(m, H2), δ 7.68(m, H2), ^195^Pt: −3150.1 ppm. **3** ^1^H NMR: δ 9.49(d, *J* = 6.03 Hz, H1), δ 7.80(merged with 12, H2), δ 8.45(t, *J* = 7.58 Hz, H1), δ 8.81(d, *J* = 8.22 Hz, H1), δ 8.31(d, *J* = 7.73 Hz, H1), δ 7.44(t, *J* = 7.85 Hz, H1), δ 7.52(t, *J* = 7.19 Hz, H1), δ 7.80(merged with 2, H2), ^195^Pt: −2217.4 ppm.

#### 3.2.2. Synthesis of [Pt(DACH)(PyPy)]^2+^ (**4a**/**b**), [Pt(DACH)(ImPy)]^2+^ (**5a**/**b**), [Pt(DACH)(BImPy)]^2+^ (**6a**/**b**) Complexes

The precursors [Pt(PyPy)(Cl)_2_] (**1**), [Pt(ImPy)(Cl)_2_] (**2**) or [Pt(BImPy)(Cl)_2_] (**3**) (1 equiv.) were stirred at 80 °C for 5 h with (1*S*,2*S*)-(+)-diaminocyclohexane (1.1 equiv.) and then left to cool to room temperature. The resulting precipitate was isolated via vacuum filtration. The precipitate was then redissolved in a 1:5 MeOH:H_2_O mixture and eluted through a 5 g C_18_ column to achieve a pure product. This method was repeated using all three [Pt(PyPy)(Cl)_2_] (**1**), [Pt(ImPy)(Cl)_2_] (**2**) or [Pt(BImPy)(Cl)_2_] (**3**) complexes previously synthesised, using (1*R*,2*R*)-(+)-diaminocyclohexane to produce 6 Pt(II) complexes in total with Cl^−^ counterions; [Pt(2-pyrrolidin-2-yl pyridine)(1*S*,2*S*-(+)-diaminocyclohexane)]^2+^ (**4a**), [Pt(2-(1H-Imidazol-2-yl)pyridine)(1*S*,2*S*-(+)-diaminocyclohexane)]^2+^ (**5a**), [Pt(2-(2-pyridyl)benzimidazole)1*S*,2*S*-(+)-diaminocyclohexane)]^2+^ (**6a**), [Pt(2-pyrrolidin-2-yl pyridine)(1*R*,2*R*-(+)-diaminocyclohexane)]^2+^ (**4b**), [Pt(2-(1H-Imidazol-2-yl)pyridine)(1*R*,2*R*-(+)-diaminocyclohexane)]^2+^ (**5b**) and, [Pt(2-(2-pyridyl)benzimidazole)(1*R*,2*R*-(+)-diaminocyclohexane)]^2+^ (**6b**). NMR spectra can be found in the Appendix A. **4a** yield = 99%, ^1^H NMR: δ 9.81(d, *J* = 5.74 Hz, H1), δ 8.69(t, *J* = 6.76 Hz, H1), δ 9.45(t, *J* = 7.70 Hz, H1), δ 9.14(d, *J* = 8.07 Hz, H1), δ 8.67(merged, *J* = 10.17 Hz, H3), δ 7.44, 7.30(t, *J* = 9.46 Hz, H2), δ 8.14, 8.03(d, *J* = 8.45 Hz, H2), δ 4.26(m, H2), δ 3.71(m, H2), δ 3.56(m, H2), δ 2.72(m, H2), δ 2.96(m, H2), δ 2.49(m, H2), ^195^Pt: −2898.1 ppm. **4b** yield = 98%, ^1^H NMR: δ 8.47(d, *J* = 5.78 Hz, H1), δ 7.36(t, *J* = 6.91 Hz, H1), δ 8.12(t, *J* = 7.76 Hz, H1), δ 7.81d, *J* = 8.13 Hz, H1), δ 7.13(merged, *J* = 9.88 Hz, H3), δ 6.12, 5.96(t, *J* = 10.41 Hz, H2), δ 6.81, 6.69(d, *J* = 7.48 Hz, H2), δ 2.93(m, H2), δ 2.36(m, H2), δ 2.02(m, H2), δ 1.38(m, H2), δ 1.62(m, H2), δ 1.16(m, H2), ^195^Pt: −2900.8 ppm. **5a** yield = 99%, ^1^H NMR: δ 8.70d, *J* = 5.73 Hz, H1), δ 7.64(t, *J* = 6.54 Hz, H1), δ 8.35(t, *J* = 7.67 Hz, H1), δ 7.40(d, *J* = 7.90 Hz, H1), δ 6.03(d, *J* = 10.20 Hz, H1), δ 7.15, 7.05(d, *J* = 7.65 Hz, H1), δ 3.20(m, merged with solvent peak), δ 2.40(m, merged with solvent peak), δ 2.06(m, H2), δ 1.43(m, H2), δ 1.58(m, H2), δ 1.15(m, H2), ^195^Pt: −2896.0 ppm. **5b** yield = 95%, ^1^H NMR: δ 8.18d, *J* = 5.69Hz, H1), δ 7.28(t, *J* = 6.63 Hz, H1), δ 8.04(t, *J* = 7.93 Hz, H1), δ 7.76(d, *J* = 7.90 Hz, H1), δ 7.25(d, *J* = 0.40 Hz, H2 = 1), δ 7.03(d, *J* = 0.25 Hz, H1), δ 2.95(m, H2), δ 2.49(m, H2), δ 2.05(m, H2), δ 1.34(m, H4 merged with 3′/6′), δ 1.64(m, H2), δ 1.18(m, H4 merged with 3′/6′), ^195^Pt: −2900.8 ppm. **6a** yield= 95%, ^1^H NMR: δ 8.67(d, *J* = 5.57 Hz, H1), δ 7.18(m, H1), δ 8.35(t, *J* = 7.88 Hz, H1), δ 8.30d, *J* = 7.39 Hz, H1), δ 7.37(d, *J* = 7.98 Hz, H1), δ 7.08, 6.90(m, H2), δ 6.61, 6.11(m, H1), δ 7.65(d, *J* = 7.83 Hz, H1), δ 2.40(m, merged with solvent peak), δ 2.12(m, H2), δ 2.02(m, H2), δ 1.46(m, H2), δ 1.60(m, H2), δ 1.18(m, H2), ^195^Pt: −2848.1 ppm. **6b** yield= 97%, ^1^H NMR: δ 8.66(d, *J* = 5.57 Hz, H1), δ 7.19(m, H1), δ 8.32(t, *J* = 7.78 Hz, H1), δ 8.32d, *J* = 7.40 Hz, H1), δ 7.37(d, *J* = 7.92 Hz, H1), δ 7.08, 6.82(m, H1), δ 6.61, 6.11(m, H1), δ 7.65(d, *J* = 7.68 Hz, H1), δ 2.42(m, merged with solvent peak), δ 2.11(m, H2), δ 2.02(m, H2), δ 1.46(m, H2), δ 1.61(m, H2), δ 1.18(m, H2), ^195^Pt: −2850.1 ppm.

### 3.3. Cytotoxicity Methodology

Calvary Mater Newcastle Hospital, Waratah, NSW, Australia, assessed the cytotoxic profile of each PC. MTT assays were performed according to methods previously published by the authors at the Calvary Mater Institute [35]. Complexes were dissolved in DMSO at high concentrations to be used as stock treatment solutions and stored at −20 °C. All cell lines used were cultured in a humidified atmosphere with 5% CO_2_ at 37 °C and maintained in Dulbecco’s Modified Eagle Medium (DMEM; Trace Biosciences, Melbourne, Australia) supplemented with 10% fetal bovine serum, sodium bicarbonate (10 mM), penicillin (100 IU·mL^−1^), streptomycin (100 μg·mL^−1^), and *L*-glutamine (4 mM). The non-cancer breast MCF10A cell line was cultured in DMEM.F12 (1:1) cell culture media. GI_50_ values were determined by plating cells in duplicate in 100 μL of medium at a density of 2500–4000 cells per well in 96-well plates. After 24 h, when cells were in logarithmic growth, media (100 μL) with or without the PC was added to each well (0 h). The growth inhibitory effects were evaluated using the MTT (3-[4,5-dimethylthiazol-2-yl]-2,5-diphenyltetrazolium bromide) assay, and absorbance was read at 540 nm after 72 h of PC exposure. The resulting absorbance data were plotted in an eight-point dose-response curve, and the drug concentration at which cell growth was inhibited by 50% (GI_50_) was calculated. These calculations were based on the difference between the optical density values at 0 h and those after 72 h of exposure to the PC.

### 3.4. Biophysical Characterization

NMR spectra were acquired using a 400 MHz Bruker Avance spectrometer (Preston/Australia). Experiments were undertaken at 298 K, using 550 µL samples prepared in DMSO. Proton (^1^H) NMR spectra were obtained using a spectral width of 8250 Hz and 65,536 data points, while Platinum (^195^Pt) NMR spectra were acquired using a spectral width of 85,470 Hz and 674 data points. ^1^H-^195^Pt HMQC spectra were recorded using a spectral width of 214,436 Hz and 256 data points for the ^195^Pt nucleus (F1 dimension) and a spectral width of 4808 Hz with 2048 data points for the ^1^H nucleus (F2 dimension). The chemical shifts of each peak were reported in parts per million (ppm), with *J* coupling reported in Hz. Spin multiplicity is reported as s (singlet), d (doublet), dd (doublet of doublet), and m (multiplet) reported in Appendix A.

UV spectra were recorded on a Cary 1E spectrophotometer at room temperature in the 190–390 nm range using a quartz cell with a 10 mm path length and an internal volume of 1 mL. All samples were corrected for solvent baseline using the Cary software (version 1.0.1284). Titration of a stock solution into a known solvent volume was used to calculate the extinction coefficient. Extinction coefficient titrations were undertaken in triplicate, and absorption data for each peak were plotted against concentration.

Fluorescent intercalator displacement (FID) assay results were obtained on a Cary Eclipse fluorescence spectrometer using quartz cells with 10 mm path length. Stock solutions of PCs **1**–**6** (30 µM) were titrated into a solution of 75 µM ethidium bromide, 150 µM ctDNA in a 40 mM K_2_HPO_4_/KH_2_PO, 10 mM KF buffer at pH 7.0. The fluorescence was measured from 550–750 nm and excited at 480 nm. Importantly, the cuvette was inverted four times, and the solution was incubated for 3 min after the addition of ctDNA to allow adequate time for the PCs to interact with the DNA before scanning. Complex concentration could then be used to calculate the stoichiometric point of binding, change in fluorescence at the point of binding (ΔF_sat_), binding coefficient (K_a_), bimolecular quenching constant (K_q_), Stern–Volmer quenching constant (K_SV_), binding constant for fluorescence (K_F_) and the molar equivalents of PC (n) at the point of intersection in the experimental stoichiometry of binding using Equations (1)–(3).
(1)$${\left[ctDNA\right]}_{T}\left[n-\frac{∆F_{x}}{∆F_{sat}}\right]=[PC]$$
(2)$$\left(\frac{∆F_{x}}{∆F_{sat}}\right)\frac{1}{n}=fraction\;of\;ctDNA-PC\;complex$$
(3)$$1-\left(\frac{∆F_{x}}{∆F_{sat}}\right)\frac{1}{n}=fraction\;free\;PC$$

The fluorescence at 601 nm of each titration was plotted, and the K_F_ and n were calculated using Equation (4).
(4)$$\frac{F_{0}}{F}=1+K_{q}\tau_{0}\left[PC\right]=1+K_{SV}[PC]$$
where F_0_ is the fluorescence of the binding site in the absence of quencher (PC), F is the fluorescence of the site containing the PC, and t_0_ is the lifetime of the chromophore in the absence of the quencher (ε_476_ = 5680 for ethidium-bound DNA). A plot of F_0_/F against [PC] using experimental values allowed the determination of K_q_ and K_SV_ from the slope as per Equation (5).
(5)$${log}_{10}\left(\frac{F_{0}-F}{F}\right)=n{log}_{10}\left[CMC\right]+{log}_{10}K_{F}$$
where n is the number of ethidium ligands that are displaced per PC. It is important to note that this expression is a simplification of the true binding interaction, as it ignores the effect of EtBr on the binding equilibrium. However, the results obtained were comparable between complexes in this study. A plot of Log_10_[(F_0_ − F)/F)] against Log_10_[PC] was used to determine K_F_ and n from the intercept and slope, respectively.

Electrospray ionization mass spectroscopy (ESI-MS) experiments were performed using a Waters TQ-MS triple quadrupole mass spectrometer in the positive mode. Sample solutions were made up to 0.5 mM in H_2_O:MeOH (90:10) and flowed at 0.1 mL·min^−1^. The desolation temperature was adjusted to 300 °C, and the flow rate of nitrogen was maintained at 500 L/h consistently throughout the measurement of all samples. Conversely, the cone voltage and capillary voltage were varied for each sample to adjust for fragmentation. Spectra were collected at 2+ *m*/*z* for each PC.

Lipophilicity was evaluated using RP-HPLC, whereby LogK_w_ is calculated by injecting a stock solution of the PC at different isocratic ratios ranging from 70–90% solvent B (organic) at a flow rate of 1 mL·min^−1^. An Agilent Technologies 1260 Infinity machine equipped with a Phenomenex Onyx™ Monolithic C_18_ reverse phase column (100 × 4.6 mm, 130 Å) was used to determine purity. The mobile phase comprised 0.06% TFA in water (solvent A) and 0.06% TFA in 90:10 ACN:H_2_O mixture (solvent B). The PC peak was eluted outside the dead zero volume time to avoid inaccurate results. The dead zero volume time was determined using potassium iodide as an external dead volume marker, with further details in the Appendix A. The subsequent peaks were recorded, and K was calculated using Equation (6).
(6)$$K=\frac{t_{r}-t_{0}}{t_{0}}$$
where K is the capacity factor, *t_r_* is the retention time of the analyte, and *t*_0_ is the dead time determined using a solution of KI in the same column. At least five different isocratic ratios were used to assess each PC, and each experiment was repeated three times. LogK’ was then calculated and plotted against the concentration of ACN in the mobile phase. The resulting linear fits of these data were then used to calculate LogK_w_, expressed by Equation (7).
(7)$$LogK^{\prime }=S\varphi +LogK_{w}$$
where *S* is the slope, *φ* is the concentration of ACN in the mobile phase, and Log*K*_w_ represents the capacity factor of the compound in 100% water (Appendix A).

CD spectra were obtained using a Jasco-810 spectropolarimeter at room temperature. The instrument was left to equilibrate for approximately 30 min, flushing the lamp with nitrogen prior to use. Spectra were obtained in a quartz cell with a 10 mm path length and were measured from 400–200 nm with a data pitch of 1 nm, bandwidth of 1 nm, and response time of 1 s. For each spectrum of each PC, 40 accumulations were collected, and a water baseline was subtracted. Additionally, the spectra were smoothed using OriginPro (version 2023B) at nine points of smoothing.

## 4. Conclusions

Six novel complexes, along with three precursors, were successfully synthesized and characterized. While the precursor complexes **1**–**3** showed significantly less DNA binding capacity and were less cytotoxic, the characterization of complexes **4**–**6a**/**b** highlighted their potential as pharmaceutical leads. The six unconventional PCs retained the chirality of the diaminocyclohexane used in their synthesis, and each demonstrated good lipophilic properties; this attribute is critical for development into effective pharmaceuticals. Although all have similar DNA binding properties, PCs **5a**/**b** and **6a** were far more cytotoxic, with GI_50_ values superior to that of Cisplatin, although by only a small margin. In contrast, complexes **4a**/**b** and **6b** were not as cytotoxic but showed far greater variation in cytotoxicity between cell lines and revealed better selectivity indices. While not as promising as possible general anticancer agents, complexes **4a**/**b** and **6b** may have success when utilized in targeted therapies. Overall, the synthesis of these PCs is uncomplicated and produces structures with favorable initial pharmaceutical properties. These promising unconventional platinum complexes can be further derivatized and tailored to help identify effective and affordable cancer treatment options.

## Data Availability

Data are available upon request to Authors.

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
