# Peer review of "Synthesis and Characterisation of Platinum(II) Diaminocyclohexane Complexes with Pyridine Derivatives as Anticancer Agents"

_ijms, 2023, doi:10.3390/ijms242417150_

Round 1

Reviewer 1 Report

Comments and Suggestions for Authors

The paper of Janice R. Aldrich-Wright and co-authors is about synthesis new Pt(II) compounds, their characterization and study of anticancer activity. The article continues the series of works by the authors in this area. I have no questions about the biological results, but I do have comments about coordination chemistry and presentation of results.

From the introduction of the article, it is not clear why the authors paid attention to diaminocyclohexane.

The authors repeatedly use the following expressions in the manuscript: “unconventional design”, “non-traditional structures”, “This non-traditional PC”, “unique design”…

What is unique about the complexes? The chemistry of platinum and platinum metals is very well studied. The Cambridge Structural Database contains 178 structurally characterized platinum complexes with diaminocyclohexane. Heteroleptic complexes with diaminocyclohexane have also been studied for quite some time. Complexes of platinum with diaminocyclohexane and nitrogen-donating ligands were obtained and characterized more than 20 years ago. I propose to remove from the text references to the uniqueness of the resulting complexes; this will not make the authors’ work lose its meaning. Examples of previously published works:

A complex of platinum with diaminocyclohexane was obtained in 1981 10.1107/S0567740881002161

Platinum complex with two diaminocyclohexane ligands and chlorine counterions published in 1977 0.3891/acta.chem.scand.31a-0182

Platinum complex with chlorine counterions with diaminohexane and another diamine characterized in 1984 https://doi.org/10.1016/S0020-1693(00)82329-4

With oxaliplatin and diaminocyclohexane in 1984 and 2003 https://doi.org/10.1023/A:1025407728854 and https://doi.org/10.1016/S0020-1693(00)80051-1

Complexes, like in the current manuscript (diaminocyclohexane + diimine) and the synthesis procedure are quite simple https://doi.org/10.1107/S0108270103020869

This 2017 article also contains the same complexes and biological research on DNA https://doi.org/10.1134/S1070328417100050

This article also outlines the methodology for the synthesis of such compounds. https://doi.org/10.1007/s11243-017-0125-0

The authors use the term “complex” to refer to the ion throughout the manuscript. This is fundamentally incorrect. Only at the end of the manuscript it is indicated that the counterions are chlorine ions. However, in earlier publications the authors did not make such mistakes. For example, here http://dx.doi.org/10.1016/j.jinorgbio.2016.06.017, the authors of the current manuscript write about counterions, which are omitted for clarity. (+ table 1, Molecular formula - The authors use the gross formula of the cation as the molecular formula. Not a complex.)

Scheme 1: The authors probably did not take Pt 4+, but K2PtCl4. Also in the scheme the reaction product is indicated not as a cation, but as a molecular complex. This needs to be corrected, charges indicated, counterions added.

Synthesis: “This contrasts with the strategy used in the synthesis of other non-conventional PCs mentioned previously, where the Pt-DACH complex is synthesised first.” - it is worth paying attention to this article and the methodology outlined long before the current manuscript was written. “An aqueous suspension of [PtCl2(bpy)] (0.33 g, 0.78 mmol) containing (1R,2R)-1,2-diaminocyclohexane (0.089 g, 0.78 mmol) was refluxed for 3 h.” http://dx.doi.org/10.1007/s11243-017-0125-0

Due to the fact that the authors did not pay attention to the counterions (Cl), the manuscript missed the key influence of chlorine anions in the action of cisplatin (see Bernhard Lippert - Cisplatin_ Chemistry and Biochemistry of a Leading Anticancer Drug-Wiley-VCH (1999), page 237). At the same time, the authors, using diaminocyclohexane, make chlorine ions more mobile, which leads to better biological properties. Pay attention to these works.

10.1021/acs.inorgchem.1c03314 “Heteroleptic Pd(II) and Pt(II) Complexes with Redox-Active Ligands: Synthesis, Structure, and Multimodal Anticancer Mechanism”

10.3390/molecules27238565 “Diimine Cisplatin Derivatives: Synthesis, Structure, Cyclic Voltammetry and Cytotoxicity”

Reaction yields need to be added to the experimental part

Good luck!

Reviewer 2 Report

Comments and Suggestions for Authors

This is a carefully performed and presented, if somewhat routine, work that adds several compounds into a large library of potentially anti-cancer Pt(II) complexes. The following minor changes are suggested:

(1) in the last sentence of the third paragraph, it should be 'anti-diabetic' not diabetic;

(2) the charges of the complexes in Figure 1 should be 2+, not +2;

(3) it is customary to include a zero line when presenting CD spectra of enantiomers, such as in Figure 2;

(4) use correct numbers of significant figures in Table 3. For example, it should be 53 +/- 1 not 53.0 +/- 1.0. This will also help to make the table more compact.

(5) Table S2 contains the same information as Table 3 and can be removed. Alternatively, Table S2 can be kept, but Table S3 can be replaced with a bar diagram.

Author Response

Reviewer 2

This is a carefully performed and presented, if somewhat routine, work that adds several compounds into a large library of potentially anti-cancer Pt(II) complexes. The following minor changes are suggested:

(1) in the last sentence of the third paragraph, it should be 'anti-diabetic' not diabetic;

Noted, we have corrected it as suggested.

(2) the charges of the complexes in Figure 1 should be 2+, not +2;

Noted, and we have corrected charges as suggested.

(3) it is customary to include a zero line when presenting CD spectra of enantiomers, such as in Figure 2;

We have added a visible “blank spectrum” line to Figure 2

(4) use correct numbers of significant figures in Table 3. For example, it should be 53 +/- 1 not 53.0 +/- 1.0. This will also help to make the table more compact.

Changes have been made as suggested. 

(5) Table S2 contains the same information as Table 3 and can be removed. Alternatively, Table S2 can be kept, but Table S3 can be replaced with a bar diagram.

We have removed Table S2